# Differentiating Polycystic Ovary Syndrome from Adrenal Disorders

**DOI:** 10.3390/diagnostics12092045

**Published:** 2022-08-24

**Authors:** Mert Yesiladali, Melis G. K. Yazici, Erkut Attar, Fahrettin Kelestimur

**Affiliations:** 1Department of Obstetrics & Gynecology, Yeditepe University Kozyatagi Hospital, Atasehir, 34854 Istanbul, Turkey or; 2Department of Endocrinology, Yeditepe University Kosuyolu Hospital, Kadikoy, 34718 Istanbul, Turkey

**Keywords:** polycystic ovary syndrome, non-classical congenital adrenal hyperplasia, Cushing’s syndrome, adrenal tumor, androgen excess disorder

## Abstract

Although polycystic ovary syndrome (PCOS) is primarily considered a hyperandrogenic disorder in women characterized by hirsutism, menstrual irregularity, and polycystic ovarian morphology, an endocrinological investigation should be performed to rule out other hyperandrogenic disorders (e.g., virilizing tumors, non-classical congenital adrenal hyperplasia (NCAH), hyperprolactinemia, and Cushing’s syndrome) to make a certain diagnosis. PCOS and androgen excess disorders share clinical features such as findings due to hyperandrogenism, findings of metabolic syndrome, and menstrual abnormalities. The diagnosis of a woman with these symptoms is generally determined based on the patient’s history and rigorous clinical examination. Therefore, distinguishing PCOS from adrenal-originated androgen excess is an indispensable step in diagnosis. In addition to an appropriate medical history and physical examination, the measurement of relevant basal hormone levels and dynamic tests are required. A dexamethasone suppression test is used routinely to make a differential diagnosis between Cushing’s syndrome and PCOS. The most important parameter for differentiating PCOS from NCAH is the measurement of basal and ACTH-stimulated 17-OH progesterone (17-OHP) when required in the early follicular period. It should be kept in mind that rapidly progressive hyperandrogenic manifestations such as hirsutism may be due to an androgen-secreting adrenocortical carcinoma. This review discusses the pathophysiology of androgen excess of both adrenal and ovarian origins; outlines the conditions which lead to androgen excess; and aims to facilitate the differential diagnosis of PCOS from certain adrenal disorders.

## 1. Introduction

Polycystic ovary syndrome (PCOS) is an androgen excess disorder that is seen in 5–20% of reproductive age women and represents the most common endocrine problem in this patient population [1,2]. The prevalence varies depending on the criteria employed; as newer, more inclusive criteria are used, the condition will be diagnosed more often [3]. Metabolic problems including type 2 diabetes, insulin resistance, and obesity are also associated with this syndrome [4]. Despite its prevalence, PCOS is still one of the most underestimated disorders among patients and physicians, with a widespread misunderstanding of the syndrome and its long-term health implications. The variations in its nomenclature, the diverse character of the disease, variations in the diagnostic criteria, and the numerous unknowns regarding its pathophysiology are all possible explanations for this situation. In addition, the mechanisms underlying androgen excess are complicated, just like androgen metabolism itself. Thus, diagnosing PCOS requires knowledge of adrenal disorders as well.

The pathophysiology and clinical presentation of PCOS are heterogenous, and the condition can be classified into numerous phenotypes based on the specific clinical pictures. Pathophysiology consists of intertwined vicious circles, but androgen excess is the culprit component of the syndrome and in the majority of patients plays the crucial role in the development of the condition. Although the ovaries are the primary source of hyperandrogenism in PCOS, up to 30% of individuals also have an excess of adrenal androgen, indicating adrenocortical hyperfunction [5]. Therefore, distinguishing PCOS from adrenal androgen excess is an indispensable step in diagnosis.

The three most recognized diagnostic criteria for PCOS are proposed by European Society of Human Reproduction and Embryology (ESHRE) together with the American Society for Reproductive Medicine (ASRM) (namely Rotterdam criteria); Androgen Excess and PCOS (AE-PCOS) Society and National Institutes of Health (NIH). All these criteria necessitate the exclusion of other androgen excess disorders and conditions that might lead to similar outcomes, like Cushing’s syndrome and non-classic congenital adrenal hyperplasia (NCAH), in particular [6,7]. The recently published International Evidence-Based Guideline for the Assessment and Management of Polycystic Ovary Syndrome, which was funded by the Australian National Health and Medical Research Council of Australia (NHMRC) and supported by a partnership with ESHRE and the ASRM, encourages the use of the Rotterdam criteria for the diagnosis of PCOS [8]. Four well-recognized PCOS phenotypes (A–D) are diagnosed according to these criteria: Phenotype A is the most common, in which all three diagnostic criteria are seen (polycystic ovarian morphology, clinical or biochemical hyperandrogenism, oligo/anovulation). Phenotype B represents patients who do not have polycystic ovarian morphology but have the other two criteria. Patients who do not have oligo/anovulation but have the other two criteria constitute phenotype C, and patients who do not have hyperandrogenism but meet the other two criteria represent phenotype D. As seen, three out of four phenotypes have clinical or biochemical androgen excess, which constitutes nearly 80% of PCOS patients [9].

This review discusses the pathophysiology of androgen excess of both adrenal and ovarian origin; outlines the conditions that lead to androgen excess; and aims to facilitate the differential diagnosis of PCOS from certain adrenal disorders.

## 2. Pathophysiology of Hyperandrogenism

### 2.1. Normal Androgen Metabolism

In order to understand the cause of androgen excess in PCOS, it is necessary to know normal androgen physiology in detail. Both the ovaries and the adrenal glands (Figure 1) produce androgens by means of their trophic hormones, LH and ACTH, via similar pathways. However, in the regulation of androgen synthesis, not only trophic hormones but also intraglandular paracrine and autocrine modulation processes appear to have a significant role as well. The steroidogenesis is explained in more detail in Section 2.2 and Section 2.3.

The main circulating androgens in women are dehydroepiandrosterone (DHEA), dehydroepiandrosterone sulfate (DHEA-S), androstenedione, testosterone, and dihydrotestosterone (in decreasing order of blood concentration). Because DHEA-S and androstenedione have little or no intrinsic androgenic activity and must be converted to testosterone to exhibit androgenic effects, they are usually regarded as pre-hormones [10]. DHEA-S is nearly entirely synthesized by the adrenals. Half of DHEA is synthesized by the adrenal cortex, nearly 20% by the ovaries, and 30% by peripheral conversion. Androstenedione is produced in almost equal amounts by the adrenal glands and ovaries [11]. Testosterone production is also shared almost equally between the ovaries and adrenals, which makes nearly half the amount of serum testosterone in healthy women. The other half of circulating testosterone is generated by peripheral conversion from androstenedione, DHEA, and estradiol (Figure 2) [12]. The majority of the serum testosterone is bound to sex hormone-binding globulin (SHBG) and albumin. Only 1–2% is free and bio-active, and this amount is highly dependent on SHBG levels and any circumstance which affects SHBG levels like estrogen levels, hyperinsulinemia, obesity, and liver conditions [13]. On the other hand, DHEA-S and androstenedione are bound to plasma proteins in negligible amounts and usually exist in plasma in free forms.

Testosterone is converted to dihydrotestosterone (DHT) in target tissues like hair follicles and external genitalia by the enzyme 5α-reductase. DHT has the highest affinity on androgen receptors and is the most potent androgen [14]. The amount of local 5-reductase activity and the concentration of androgen receptors modulate the extent of the androgenic effects of testosterone in target tissues. Enzymes like oxidative 17β-hydroxysteroid dehydrogenase (17β-HSD) and aromatase that convert testosterone into ineffective androgens (androstenedione) and estradiol are also found in target tissues. It seems that there is a local regulatory mechanism for androgenic activity in target tissues, and disruption of this balance is pivotal in some androgen excess disorders.

In females, the neuroendocrine system has no direct feedback control over androgen production. Intraglandular paracrine and autocrine processes appear to have a vital role in the regulation of androgen synthesis. Furthermore, local androgen concentrations in ovaries affect steroidogenesis and follicular growth [15]. Androgens are crucial substrates for the synthesis of estradiol, but in higher local concentrations, they suppress ovulation. Hence, intraovarian mechanisms appear to modulate the ovarian androgenic response to LH in order to regulate androgen and estrogen production and enhance follicular development. In the ovaries and adrenal glands, androstenedione is the primary precursor for both testosterone and estrogen. In granulosa cells, androstenedione is 10-fold more abundant than testosterone, and thus, it is the major aromatase substrate for estradiol formation [16].

Some extraovarian modulators such as insulin have the ability to overcome natural intraovarian down-regulatory processes that regulate ovarian androgen production. The presence of insulin receptors on the membrane of follicular cells indicates that insulin can modulate theca cells directly to elicit hormone biosynthesis [17]. It has been shown that insulin increases androstenedione production in theca cells, activating the production of StAR and CYP17A1 mRNA, which increases androgen levels [18,19]. Insulin also increases CYP17 and p450scc levels in a synergistic manner with human chorionic gonadotrophin (HCG), resulting in increased androgen production [20].

In terms of steroid production, the zona reticularis in the adrenal gland is similar to the theca cells in the ovary. Under normal circumstances, theca cells in ovaries mostly produce DHEA and androstenedione, with a lesser amount of testosterone. At the time of ovulation, the normal increase of stromal tissue causes a spike in androstenedione and testosterone levels. In case of an androgen-secreting tumor or increased stromal tissue, testosterone secretion becomes prominent [21]. Although the pathways of steroidogenesis are similar, there are certain differences in androgen excess disorders of adrenal origin and ovarian origin, as will be discussed later.

### 2.2. Adrenal Steroidogenesis and Androgen Excess of Adrenal Origin

The adrenal glands are extremely complicated in both function and structure. They consist of an adrenal cortex divided into three distinct zones and an adrenal medulla. The cortex leads to the synthesis of adrenal androgens in the zona reticularis, mineralocorticoids in the zona glomerulosa, and glucocorticoids in the zona fasciculata. The renin–angiotensin–aldosterone system regulates the activity of the zona glomerulosa enzymes, whereas the hypothalamic–pituitary–adrenal (HPA) axis regulates two additional major pathways. The adrenocorticotrophic hormone (ACTH) promotes the release of cortisol in the zona fasciculata and adrenal androgens in the zona reticularis [22].

Adrenal androgen precursors are released from the zona reticularis by the action of pituitary ACTH. The activity of 3-hydroxysteroid dehydrogenase (3β-HSD) type 2 on the 3-beta-hydroxyl group transforms the majority of DHEA to androstenedione. DHEA is synthesized from cholesterol and pregnenolone by the theca cells’ cholesterol side chain cleavage enzyme and 17-hydroxylase. DHEA may then be changed to androstenedione and testosterone [16]. Androstenedione is converted to 11β-hydroxyandrostenedione (11OHA4), the 11-oxygenated androgen precursor, by the enzyme cytochrome P450 11β-hydroxylase type 1 (CYP11B1) in the adrenal gland [23]. It is subsequently transformed to 11-ketoandrostenedione (11KA4) in the kidney by 11-hydroxysteroid dehydrogenase type 2 (11HSD2). In peripheral tissues, 11KA4 is activated by AKR1C3 to produce 11-ketotestosterone (11KT); 11-oxygenated androgens account for the bulk of the circulating androgens in PCOS, early adrenarche, and congenital adrenal hyperplasia (CAH). 11KT and 11-ketodihydrotestosterone (11KDHT) are potent androgens that bind to androgen receptors with high affinity, such as testosterone and DHT [24].

Androgenic precursors such as DHEA and androstenedione may be activated into more powerful active androgens such as testosterone and dihydrotestosterone in the periphery depending on the presence of androgen-activating enzymes in local tissues [25]. 3β-HSD 1 and 2 are found in the adipose tissue, liver, brain, skin, and breast and may convert DHEA to androstenedione. DHEA and DHEA-S are primarily metabolized in the liver by reduction and conjugation to sulphates or glucuronides; 17-ketosteroids are excreted in the urine as metabolites of these steroids and androstenedione.

While DHEA and DHEA-S are released in modest levels throughout infancy and early childhood, they gradually increase in late childhood and are related to the growth of pubic and/or axillary hair in prepubertal children during a period known as adrenarche. The cause of this rise in androgen secretion is unclear; nevertheless, it coincides with the formation of the adrenal cortex’s zona reticularis. The lack of increased ACTH and cortisol secretion shows that ACTH is not responsible.

Excessive androgen production in the adrenal cortex can develop from both acquired and hereditary adrenal disorders, as well as other conditions that impact adrenal function (Table 1).

### 2.3. Androgen Excess Mechanism in PCOS

Although polycystic ovarian morphology was once considered to be responsible for anovulation, it is now accepted as a consequence of chronic anovulation, in which androgen excess is thought to be a culprit. Women with PCOS produce more androgens and estrogens on a daily basis, as seen by higher plasma levels of DHEA, DHEA-S, androstenedione, testosterone, and estrone. Patients with PCOS have higher blood LH concentrations, lower or normal FSH levels, and thus higher LH: FSH ratios than women who are ovulatory [26]. There is an increase in LH frequency and amplitude which is manifested by an increase in LH serum values [27] and which directly affects ovarian androgen secretion. An increase in GnRH pulsatility is also seen in PCOS patients, which causes a decrease in FSH serum levels and contributes to elevated LH levels. Increased estrone concentrations produced by the aromatization of excess androstenedione and elevated inhibin B produced in small antral follicles may also contribute to lower FSH levels [28].

**Ovarian contribution:** The etiology of increased ovarian androgen production in PCOS is complex and cannot be reduced to a single factor. Increased LH stimulation as a result of aberrant LH secretory dynamics is a significant factor [27]. Increased theca cell volume and increased theca cell sensitivity to LH stimulation are also among possible reasons for increased ovarian androgen production [29]. Androgen production remains elevated in theca cell cultures obtained from PCOS patients, which suggests an intrinsic dysregulation of steroidogenic enzymes in ovarian theca cells [30,31]. The dysregulation of the cytochrome p450c17α is characterized by a high 17-hydroxyprogesterone (17-OHP) response to GnRHa testing through 17-hydroxylation and increased but relatively inefficient activity of 17,20-lyase and may be responsible for increased androgen levels seen in patients with PCOS [32,33]. It has also been suggested that LH biological activity may differ among people as a result of polymorphism in the gene encoding beta subunit of the hormone [34]. In addition, theca cells of PCOS follicles were found to express higher LH receptors and CYP11A and CYP17 mRNAs than theca cells of same-size control follicles [35]. Hyperinsulinemia also potentiates LH action in ovaries, which is mentioned below.

**Adrenal Contribution:** Although the source of androgen excess is mostly the ovaries, more than 50% of the PCOS patients also have increased androgen production from the adrenals, which is manifested as increased serum DHEAS levels [36]. Adrenal androgens have minimal or no androgenic activity, but they do play a role in the pathogenesis of hyperandrogenism through the peripheral conversion to testosterone. Although the etiology of the increase in adrenal androgen production is not clear, various mechanisms have been investigated so far. Nearly half of PCOS patients have an increased androgen production response to exogenous ACTH administration [36]. In both males and females, adrenal androgen secretion diminishes with aging; however, according to the findings of one study, women with PCOS had excessive adrenal androgen release with no decline as they approached menopause [37]. Chronically elevated estrogen levels due to anovulation could decrease adrenal 3-HSD activity, but the evidence for this mechanism is not conclusive. It has also been suggested that in some PCOS patients, increased adrenal androgen may be due to an increase in intrinsic P450c17 17,20-lyase activity [38]. On the other hand, it has been reported that there is an alternate pathway in the adrenals in women with PCOS, which causes an increase in adrenal androgens [39].

Insulin Resistance: Insulin resistance itself, which is seen in 50–75% of PCOS patients, appears to be a mechanism that can lead to androgen excess [40]. Androgen excess is caused or exacerbated by insulin resistance in at least two ways: by promoting increased ovarian androgen production and by suppressing hepatic SHBG synthesis. Numerous studies have shown that insulin promotes androgen synthesis in ovarian theca cells in vitro, and theca cells in PCOS women have increased insulin sensitivity [41,42]. It has also been demonstrated that insulin potentiates the activity of LH, implying that insulin and LH work synergistically to increase androgen synthesis [43]. Serum testosterone concentrations decline when insulin resistance is treated with metformin [44] or insulin levels are reduced with diazoxide [45]. Insulin and androgens work synergistically to reduce SHBG levels, resulting in higher free androgen levels, which aggravates the insulin resistance [46]. Hence, insulin resistance and androgen excess in PCOS form a vicious circle that gradually increases the severity of the condition.

Anti-Müllerian hormone (AMH), which is usually elevated in PCOS patients, has also been held to account for contributing to PCOS pathophysiology. AMH is secreted primarily from pre-antral and small antral follicles, which are increased in number in PCOS patients. Early-stage follicle development has been demonstrated to be inhibited by AMH [47], and the reduction in granulosa cell sensitivity to FSH caused by AMH was also validated in vivo [48]. Furthermore, it has been shown to decrease aromatase enzyme activity, thereby inhibiting estradiol production and contributing to the androgen excess in PCOS [49].

As mentioned earlier in Section 2.1, in healthy women, testosterone and androstenedione are produced nearly in equal amounts by the ovaries and adrenals. However, in women with PCOS, these two androgens are predominantly produced by ovaries with a lesser contribution from adrenals (Table 2). Androstenedione and testosterone are both produced nearly 60% by ovaries, rest of testosterone is mostly produced by peripheral conversion, and the rest of androstenedione is mostly produced in the adrenals [50]. As the biologic effect of testosterone is determined by its free fraction, SHBG level, which is usually altered in PCOS patients, is crucial. Androgens and insulin, which both decrease the liver production of SHBG, are usually raised in PCOS patients [51].

In healthy women, androstenedione is secreted by LH stimulation in theca cells in significant quantities and is converted to estrogens in granulosa cells. It also acts as a precursor of testosterone and is converted to testosterone by the enzyme 17-beta-hydroxysteroid dehydrogenase, which is found in most tissues. Since LH levels are higher in PCOS patients, there is an increased secretion of androstenedione from ovarian theca cells. Furthermore, theca cells from PCOS women are more responsive to LH-stimulated androgen synthesis, which contributes to androstenedione excess [29]. Some part of this excess androstenedione undergoes peripheral conversion to testosterone by the enzyme 17-beta-hydroxysteroid dehydrogenase.

Since DHEAS is mostly produced in adrenals, the serum level of DHEAS is a good indicator of adrenal-derived androgen excess. However, some adrenal carcinomas have no sulfotransferase, which is the enzyme that converts DHEA to DHEAS. In addition, DHEAS levels are not always elevated in congenital adrenal hyperplasia patients. Hence, normal DHEAS levels are not always enough to rule out adrenal pathologies, but elevated levels strongly suggest adrenal originated androgen excess [52].

## 3. Differential Diagnosis

Taking a broad view, while in adrenal androgen excess, serum DHEAS level is usually elevated, high serum testosterone levels usually suggest ovarian androgen excess. In idiopathic hirsutism or PCOS, serum DHEAS concentration is normal or slightly raised. The differential diagnosis of the most common adrenal androgen disorders and PCOS is discussed in detail.

### 3.1. Non-Classical Congenital Adrenal Hyperplasia (NCAH)

Both PCOS and NCAH are hyperandrogenic disorders that require thorough differential diagnosis in teenage girls and older women [53]. Despite the fact that physicians see young women with acne, hirsutism, or both, NCAH or PCOS may not be fully validated.

Non-classical congenital adrenal hyperplasia (NCAH) is usually seen as a result of 21 hydroxylase deficiency (21 OHD). Its prevalence in the Caucasian race is about 1/200, and it shows autosomal recessive inheritance. The most common mutation is CYP21A2. Other mutations can be listed from mildest to most severe as V281L, P30L, I172N, I2 splice, and null. The carrier frequency for a severe NCAH mutation is approximately 1/60, while mild mutations ranging from 1/5 to 1/16 of the population have been detected. For this reason, it is important that even heterozygous mutation carrier couples who do not need medical treatment receive genetic counseling before pregnancy.

In classical adrenal hyperplasia cases diagnosed with ambiguous genitalia in the neonatal period, 21 hydroxylase enzyme activity was 1%; it is preserved in 20–50% of cases of NCAH and is often not diagnosed until adolescence. While ACTH levels are elevated in a few patients, as in classical CAD, ACTH levels are generally within normal limits in NCAH [54]. The response of cortisol to ACTH is variable: Normal or mild impairment may occur, and some patients may experience hypersensitivity. The androgen response to ACTH from the adrenal glands increases; however, DHEAS levels are usually normal, while DHT, testosterone, and androstenedione are high [54,55,56].

While it is marked that children with NCAH have increased stature and advanced bone age [57], studies have shown that these girls enter puberty on average 6 months earlier [55,58,59,60]. Although the clinical presentation varies from patient to patient, usually, symptoms of hyperandrogenemia such as hirsutism, frontal/temporal alopecia, acne, and oily skin in addition to menstrual disorders [55,60,61] and clitoromegaly [60] are noted during adolescence.

Clinically, PCOS and NCAH have many common symptoms and are difficult to distinguish with only physical examination and pelvic ultrasound. The presence of clitoromegaly in the population with NCAH, unlike PCOS, is marked and is an important finding for differential diagnosis [62]. However mean age, BMI, waist/hip ratio, hirsutism scores, and acne prevalence were found to be similar in patient populations with NCAH and PCOS [63]. Androgenic alopecia, menstrual and ovulation disorders, onset of puberty, and positive family history are also common in women with either NCAH or PCOS [55]. In addition, the ultrasound finding (polycystic ovarian morphology, PCOM) observed in 75% of the PCOS population is also seen in 40% of the NCAH population, especially in the ovulation disorder subgroup, and does not help in the differential diagnosis. Hence, the sole approach for distinguishing NCAH patients from PCOS patients is to assess 17-OHP levels [64]. Traditionally in these patients, if the morning 17-OHP level is above 2 ng/mL in the early follicular phase, an ACTH stimulation test is performed. Tetracosactrin (1–24) is used for ACTH stimulation testing. Blood samples for the measurement of 17-OHP and cortisol, if required, are obtained in the basal state and 30 and 60 min after the administration of 250 microgram intravenous ACTH [60,65]. A 17-OHP value over 10 ng/mL during the test indicates NCAH regardless of mutation in genes [66]. Recently, a cut-off value of 5.4 ng/mL has been proposed in the differential diagnosis of NCAH and PCOS, dismissing the ACTH stimulation test [66].

### 3.2. Cushing’s Syndrome

Cushing’s syndrome (CS) is defined as an excess of glucocorticoids in the body. It encompasses a variety of clinical symptoms triggered by the chronic overproduction of endogenous glucocorticoids or by long-term steroid or ACTH treatments [67,68]. Endogenous Cushing’s syndrome usually has one of two mechanisms: ACTH dependent or ACTH independent. ACTH-secreting pituitary adenoma, known as Cushing’s disease (CD), constitutes 60–70% of cases with Cushing’s syndrome. In 30 to 40% of cases, Cushing’s syndrome is independent of ACTH, due either to ectopic ACTH production or an adrenocortical tumor. Another cause of ACTH-independent Cushing’s syndrome is primary adrenocortical nodular dysplasia with atrophy of the adjacent adrenocortical areas.

The phenotype of Cushing’s Syndrome varies from patient to patient. The reason for this is the differences in the degrees of hypercortisolism in patients and the sensitivity difference in the glucocorticoid receptors [69,70]. Typical symptoms of Cushing’s syndrome are supraclavicular fat deposition; central obesity with a full moon face and buffalo hump on the neck; typical lilac-purple streaks in the folds of the abdomen, thighs, and axillae; and easy bruising and thinning of the skin. Fatigue and muscle weakness, proximal myopathy, depressed mood, sleep disturbances, anxiety, hypertension, glucose intolerance, menstrual cycle disorders, hirsutism, and acne are other manifestations of CS. Significant virilization may be indicative of an adrenocortical carcinoma [71].

Menstrual irregularity is seen in 70–80% of the female population with CS, and 46% of these patients have signs of PCOS. Elevated cortisol in the blood has an effect on the hypothalamus, suppressing the synthesis and secretion of gonadotropin-releasing hormone, thus inhibiting the secretion of LH and FSH hormones. Low estradiol levels, excessive androgen secretion from the adrenal gland, and a slight increase in prolactin levels may also be observed in some patients. SHBG levels in the blood lower due to decreased production in the liver. Because of increased androgen bioavailability, hirsutism develops even when androgen levels are at normal levels [70,72]. Menstrual irregularity and amenorrhea were associated with higher cortisol levels rather than estradiol levels in these women [73].

The clinical course of Cushing’s disease is often considerably more indolent than that of hypercortisolism caused by an adrenocortical carcinoma (ACC), and it is frequently misdiagnosed as PCOS. In a study, it was reported that half of the patients with CD were initially diagnosed with PCOS. Hirsutism and menstrual irregularities were more common among patients with CS, who were initially diagnosed with PCOS [74]. In CS, signs of hypercortisolism predominate, while in PCOS, signs of hyperandrogenism predominate the clinical situation. Among the clinical findings, hirsutism, obesity, acne, alopecia, striae, menstrual irregularities, insulin resistance, and depression are findings that are common to CS and PCOS. However, there are some distinctive clinical features that should be kept in mind. For example, the stria in CS are atrophic and purple in color, while those in PCOS are usually narrow and pale. Another distinctive clinical feature is that in CS patients, proximal myopathy, easy bruising, and thin skin are encountered, while PCOS patients usually have thick skin and good muscle mass (Figure 3).

After the clinical examination, the next steps are endocrinological function tests and imaging. Prior to biochemical testing, patients should be questioned about alcohol abuse or exogenous glucocorticoid therapy (oral, parenteral, rectal, inhaled, or topical). First-line screening tests for the diagnosis of hypercortisolism are 24-h urine cortisol measurement and/or low-dose dexamethasone suppression test (DST) and/or late-night salivary cortisol measurement. For the dexamethasone suppression test, a 1 mg dexamethasone tablet is taken between 11 pm and midnight, and blood is drawn for cortisol at 8 am the next morning. Cushing’s syndrome is excluded if the serum cortisol level is <1.8 mg/dL. Another type of dexamethasone suppression test is a 2 mg, 2-day test; dexamethasone 0.5 mg is given every 6 h for two days, and serum cortisol level is measured early in the morning (9 a.m.) after the last dose of dexamethasone [75]. An alternative test is multiple salivary cortisol measurements taken at 23:00. However, it should not be forgotten that sleep disorders cause physiological hypercortisolism and that this test can only be performed in patients without sleep disorders. Noninvasive tests used in the differential diagnosis of CS are ACTH level measurement, corticotrope-releasing hormone (CRH) stimulation test, high-dose DST, and appropriate imaging (pituitary MRI and adrenal CT or MRI) [70,72,76].

### 3.3. Androgen Secreting Adrenal Tumors

Although both benign and malignant adrenal tumors can secrete androgens, androgen-secreting adrenal tumors are usually malignant. Among women with androgen excess, benign androgen-secreting adrenal adenomas are rarely seen, and they usually do not cause significant hyperandrogenemia or hyperandrogenism. If serum androgen levels are high, they do not respond to dexamethasone [74]. Androgen-secreting adrenal adenomas are generally small (<4 cm) in size.

In a patient with hyperandrogenism, rapidly developing androgenic symptoms should raise suspicion for an androgen-secreting adrenal malignant tumor rather than PCOS. These patients usually have very high DHEA and DHEAS levels (DHEAS: >700 ng/mL; 19 μmol/L). Even though the predictive value is limited, very high serum testosterone levels (>200 ng/dL; 6.9 nmol/L) should also raise suspicion for androgen-secreting adrenal tumors [77]. Weight loss, flank pain, and back pain are all signs that suggest adrenal tumors. Blood cortisol levels and cortisol excretion in the urine are usually increased. Malignant tumors are usually bigger than adenomas (>5 cm) [78]. An abdominal computed tomography (CT) or magnetic resonance imaging (MRI) scan should be performed if suspicion is high.

## 4. Discussion

Hirsutism is the most common presenting symptom of patients with androgen excess disorder; it affects 5–8% of the female reproductive age population [53,65,79]. PCOS accounts for the majority of the women who seek medical advice because of hirsutism. Although PCOS is the most prevalent cause of hyperandrogenism, it is important to remember that other diseases can also cause it. Many illnesses, including NCAH, idiopathic hyperandrogenemia (IHA), Cushing’s syndrome, acromegaly, adrenal ovarian tumors, and various medicines produce androgen excess. The presence of at least two of the three Rotterdam criteria, as well as the exclusion of other illnesses or conditions that may mimic polycystic ovarian syndrome, is sufficient for the diagnosis of PCOS [80]. In women of reproductive age or hyperandrogenic women, PCOS is seen 40–50 times more commonly than NCAH [64].

The activity of the 21-hydroxylase enzyme might vary between 20 and 75% as a consequence of mutations in the CYP21A2 gene. Decreased enzymatic activity secondary to this activity also leads to increased adrenocorticotropic hormone (ACTH) stimulation with the accumulation of cortisol precursors that play a role in androgen biosynthesis. Although some parameters such as obesity, insulin resistance, luteinizing/follicle stimulating hormone ratio (LH/FSH), and polycystic ovarian morphology are observed more frequently in PCOS patients, they cannot be used in differential diagnosis since the two diseases may appear in similar clinical pictures. In addition, it has been determined that serum basal 17-OHP, androstenedione, and testosterone levels may be high in both PCOS and NCAH. However, serum basal 17OHP levels are generally significantly higher in NCAH. For this reason, follicular phase 17OHP levels are used as the basic screening tool in the differential diagnosis.

Genetic testing can be used as an alternative NCAH diagnostic tool when biochemical results are uncertain or pre-pregnancy genetic counseling is required [81]. In particular, CYP21A2 genotyping should be performed to identify heterozygous gene carriers. On the other hand, there is no specific genetic test to determine the risk of developing PCOS in female offspring or to diagnose PCOS.

Cushing’s disease is rare in women who develop symptoms of hyperandrogenism within a short period of time. However, it should be excluded in a patient with accompanying symptoms of hypercortisolism. Diagnosis can be made by performing a dexamethasone suppression test or by measuring free cortisol levels in urine collected for 24 h [82]. In the very rare case of severe virilization, a differential diagnosis of androgen-secreting tumors of ovarian or adrenal origin should be made. Tumors that secrete androgen at very high levels should definitely be considered in the foreground. An algorithm for the differential diagnosis of hyperandrogenism is demonstrated in Figure 4.

## 5. Conclusions

In conclusion, PCOS and adrenal disorders characterized by androgen excess lead to similar clinical pictures. Before the diagnosis of PCOS is made, it is necessary to exclude hyperandrogenaemia due to adrenocortical disease. Medical history and physical examination are very helpful in the differential diagnosis between PCOS and Cushing’s disease in particular. There may be clinical difficulties in distinguishing between NCAH and PCOS. The most important parameter in the differential diagnosis is the basal and ACTH stimulated 17-OH progesterone level when required in the early follicular period. It should be kept in mind that rapidly progressive hyperandrogenic manifestations such as hirsutism may be due to an androgen-secreting adrenocortical carcinoma.

## Figures and Tables

**Figure 1 diagnostics-12-02045-f001:**
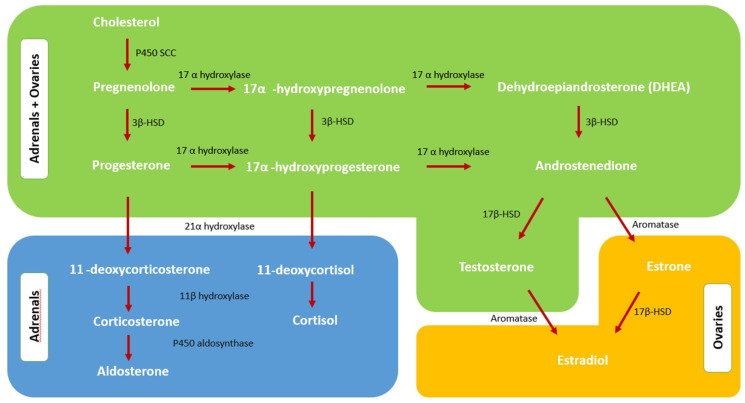
Steroidogenesis pathways in adrenals and ovaries. P450 SCC: side chain cleavage enzyme, 3β-HSD: 3β-Hydroxysteroid dehydrogenase, 17β-HSD: 17 beta hydroxysteroid dehydrogenase.

**Figure 2 diagnostics-12-02045-f002:**
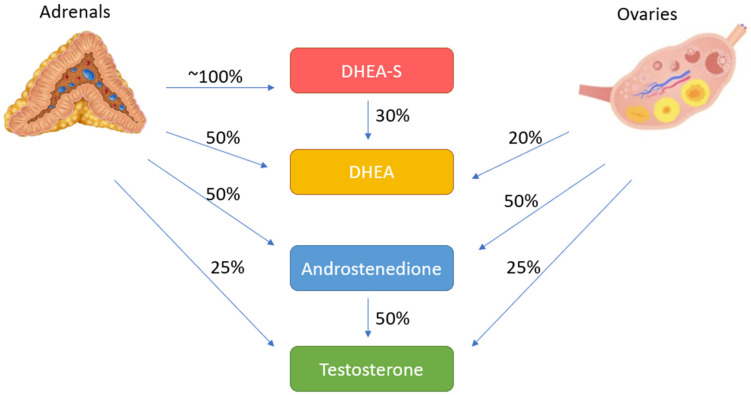
Sources of serum androgens in healthy women.

**Figure 3 diagnostics-12-02045-f003:**
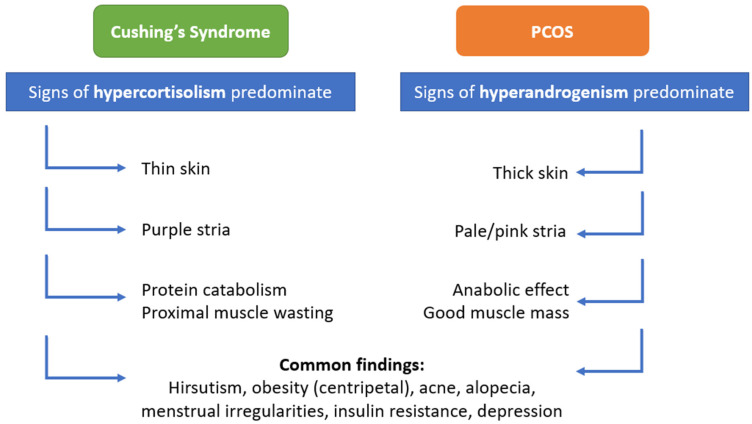
Clinical findings in CS and PCOS.

**Figure 4 diagnostics-12-02045-f004:**
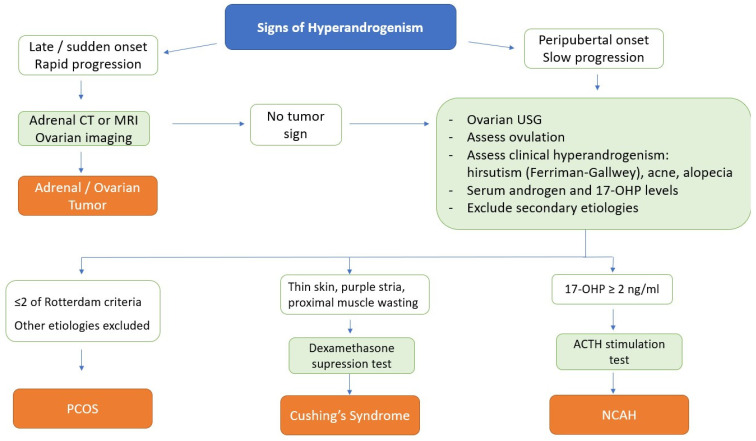
A differential diagnosis algorithm for hyperandrogenism.

**Table 1 diagnostics-12-02045-t001:** Adrenal Originated Androgen Excess Causes.

Primary Adrenal Diseases	ACTH Hypersecretion	Pregnancy	Other Causes
Premature adrenarche Adrenal tumors	Congenital adrenal hyperplasia ACTH-dependent Cushing’s syndrome Glucocorticoid resistance Cortisone reductase deficiency	Placental sulfatase deficiency (no hyperandrogenism) Placental aromatase deficiency (mother and female fetus virilization) P450 oxidoreductase deficiency	Hyperprolactinemia Exogen DHEA intake PAPSS2 deficiency

**Table 2 diagnostics-12-02045-t002:** Sources of serum androgens in PCOS patients.

Androgen	Ovary	Adrenal	Peripheral Conversion
Testosterone	60%	5%	35% (from androstenedione)
Androstenedione	60%	35%	5% (from DHEAS)
DHEAS	<5%	>95%	0%

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
