# Peer review of "Differentiating Polycystic Ovary Syndrome from Adrenal Disorders"

_diagnostics, 2022, doi:10.3390/diagnostics12092045_

Round 1

Reviewer 1 Report

The paper is well written and falls within the scope of the journal. I have appreciated the presence of figures.

I have only minor concerns:

In your opinion, how the different PCOS phenotypes could help or delay the right diagnosis?

In order to improve your reference, I advise to take into account these two papers for you introduction/discussion section: doi: 10.1080/09513590.2017.1391205; doi: 10.1080/13543784.2020.1781815.

Author Response

1-In your opinion, how the different PCOS phenotypes could help or delay the right diagnosis?

The reviewer brings up an important point. In our opinion, the phenotype including PCOM and menstrual abnormalities/anovulation delays, the phenotype including hirsutism, high androgen levels and oligo/amenorrhea helps the diagnosis of PCOS. I think that this issue may not needs to be mentioned in the manuscript.

2-In order to improve your reference, I advise to take into account these two papers for you introduction/discussion section: doi…

These two references were added

Reviewer 2 Report

Dear Authors,

The presented study tackles an important issue of Differentiating Polycystic Ovary Syndrome from Adrenal Disorders. I have read the manuscript with great interest. The study was conducted reliably with the appropriate selection of articles.

However, some issues require complementary information:

1.       I suggest redrafting  the Abstract. The Abstract should be a reume of the manuscript not the Introduction. I suugest including major information about differentiating of PCOS and Adrenal Disorders.

2.       I suggest including a figure to show differential diagnosis more clear.

3.       I suggest including some more information about the procedurÄ™ of ACTH-stimulation test.

4.       I suggest including the information about 2 mg dexametasone test.

5.       I suggest including Discussion section.

Author Response

1-I suggest redrafting the abstract. The abstract should be resume of the manuscript not for the introduction. I suggest including major information about differentiating of PCOS and adrenal disorders.

We appreciate the reviewer for this comment. Abstract was rewritten.

2-I suggest including a figure to show differential diagnosis more clear.

Thanks for this suggestion. We have included a figure to show differential diagnosis more clear.

3-I suggest including some more information abot the procedure of ACTH stimulation test.

The procedure of ACTH stimulation test was stated in the relevant section.

4-I suggest including the information about 2 mg dexamethasone test

The procedure of 2 mg dexamethasone suppression test was also described in the relevant section.

5-I suggest including discussion section.

A discussion section was added.